# Systems Architecture Design Pattern Catalog for Developing Digital Twins

**DOI:** 10.3390/s20185103

**Published:** 2020-09-07

**Authors:** Bedir Tekinerdogan, Cor Verdouw

**Affiliations:** 1Information Technology Group, Wageningen University and Research, P.O. Box 35, 6700 AA Wageningen, The Netherlands; cor.verdouw@wur.nl; 2Mprise, P.O. Box 598, 3900 AN Veenendaal, The Netherlands

**Keywords:** digital twins, system engineering, system architecture design, smart agriculture, internet of things, farm management systems, remote sensing and control, virtualization

## Abstract

A digital twin is a digital replica of a physical entity to which it is remotely connected. A digital twin can provide a rich representation of the corresponding physical entity and enables sophisticated control for various purposes. Although the concept of the digital twin is largely known, designing digital twins based systems has not yet been fully explored. In practice, digital twins can be applied in different ways leading to different architectural designs. To guide the architecture design process, we provide a pattern-oriented approach for architecting digital twin-based systems. To this end, we propose a catalog of digital twin architecture design patterns that can be reused in the broad context of systems engineering. The patterns support the various phases in the systems engineering life cycle process, and are described using a well-defined pattern documentation template. For illustrating the application of digital twin patterns, we adopt a multi-case study approach in the agriculture and food domain.

## 1. Introduction

A digital twin refers to a digital replica of potential or actual entities (i.e., physical twin) [1,2]. It provides rich representations of the corresponding physical entity and enables sophisticated control for various purposes. A key characteristic of a digital twin is that it is connected to a physical entity which is typically established by the use of real-time data using sensors. Digital twins are made possible through the integration of various technologies such as Internet of Things [3,4,5], artificial intelligence, machine learning, and data science, which enable living digital simulation models to be created that reflect the changes of the physical counterparts. Hereby, a digital twin continuously learns and updates itself using data from sensors or external entities. The key motivations for adopting digital twins are to be able to design, test, manufacture, and use the virtual version of the systems [1]. In the industrial sector, for example, digital twins can be used to optimize the operation and maintenance of physical assets and manufacturing processes. Overall, digital twins can be applied to a broad range of application domains, including manufacturing, aerospace, smart farming, healthcare, and the automotive industry.

Although the concept of digital twins has received increased interest, designing digital twins has not yet been fully explored. The design of digital twin-based systems involves consideration of a physical entity, a virtual entity, and the connection among these. Because the concept consists of a view of both virtual and physical elements, we can consider this as a systems engineering approach that adopts a holistic perspective for developing a system to meet multidisciplinary stakeholder goals. An important asset in systems engineering is the system architecture that defines the gross level, systemic structure of the system. The system architecture defines the gross-level, systemic structure, behavior, and views of a system, and is typically documented using a formal description and representation of a system, organized in a manner that supports reasoning about the structures and behaviors of the system. The system architecture consists of system components and the sub-systems that work together to implement the overall system. In a digital twin-based system, the system architecture includes the two key components of the digital twin and the physical twin.

Depending on the context and overall purpose, in practice, digital twins can be applied in different ways, thus leading to various architectural designs. Designing an architecture from scratch is not easy and requires a thorough background of both the application domain and the architecture design and documentation principles. To leverage design expertise and support reuse, the software and systems engineering community have introduced design patterns that can be applied at different levels in the system development lifecycle, including the architecture design, detailed design, and the code. A design pattern is a documented best practice and generic solution to recurring problems.

Currently, designers of digital twin-based systems appear to rely on informal design practices to describe the architectures of the configurations of digital twin components that make up the overall system. To guide the architecture design process and leverage reuse, we provide a pattern-oriented approach for architecting digital twin-based systems. To this end, we propose a set of digital twin architecture design patterns that can be reused to design the architecture of a system that meets the corresponding digital twin requirements. The patterns are described using the well-known pattern templates of context–problem–solution. For illustrating the application of the digital twin patterns, we adopt a case study in the agriculture and food domain derived from the European IoF2020 project.

The remainder of this paper is organized as follows. In Section 2 we provide the background of digital twins and architecture design. Section 3 presents the research methodology for discovering and modeling the digital twin pattern catalog. Section 4 presents the domain analysis process for digital twins that is used as a basis for describing the patterns. Section 5 presents the digital twin pattern catalog. Section 6 presents the discussion, and finally, Section 7 concludes the paper.

## 2. Background and Key Concepts

Developing architecture design patterns requires an understanding of several independent but related domains, including systems engineering, architecture design, and architecture design patterns. We provide the background on these in the following sub-sections.

### 2.1. Systems Engineering

Systems engineering is a broadly adopted concept that includes the conception, design, development, production, and operation of physical systems. According to International Organization for Standardization and the International Electrotechnical Commission (ISO/IEC 42010) [6], the notion of a “system” is defined as a set of components that accomplishes a specific function or set of functions. The notion of *systems engineering* is defined as “an interdisciplinary approach to translating users’ needs into the definition of a system, its architecture and design through an iterative process that results in an effective operational system. Systems engineering applies over the entire life cycle, from concept development to final disposal” [7,8].

In general systems are not standalone and interact with its environment, which may include other systems, users, and the natural environment. A further characteristic of a system is that it is usually composed of a variety of system elements including hardware, software, firmware, people, information, techniques, facilities, services, and other support elements [7,8]. The person or role who supports this transdisciplinary approach and applies the systems engineering process is defined as a *systems engineer*. In particular, the systems engineer elicits and translates customer needs into specifications that can be realized by the system development team. As such, a systems engineer aims to ensure the elements of the system fit together to accomplish the objectives of the whole, and ultimately satisfy the needs of the customers and other stakeholders who will acquire and use the system. A system is developed for or impact different stakeholders. A stakeholder is defined as an individual, team, or organization with interests in, or concerns relative to, a system. A concern is defined as a matter of interest in the system that could be functional or related to quality issues. Architectural drivers define the concerns of the stakeholders that shape the architecture.

To ensure the development of successful systems that meet the stakeholder concerns, systems engineering is performed based on a systematic life cycle process beginning early in conceptual design and continuing throughout the life cycle of the system through its manufacture, deployment, use, and disposal. A systems engineering (SE) process defines the primary activities that must be performed to implement systems engineering. Figure 1 shows the generic ISO/IEC Systems Lifecycle [7] including the key stages of the systems engineering process. The *concept stage* identifies and explores the stakeholders’ needs and the enabling technologies to provide a high-level, preliminary concept for the system. The *development stage* elaborates on the selected concept, and defines and realizes a system of interest (SOI). The resulting SOI should meet the stakeholder requirements and can be produced, utilized, supported, and retired. The conceptual representation of the systems architecture is developed in the concept stage and is further elaborated and designed in the development stage. The system is produced or manufactured during the *production stage*. During this stage, product modifications may still be required to resolve production problems, reduce production costs, or enhance product or system capabilities. The *utilization stage* is the stage in which the system is operated in its intended environment to deliver its intended services. The *support stage* is the stage in which the system provides services that enable continued operation. The *retirement stage* is the stage in which the system and its related services are removed from operation.

### 2.2. Architecture Design

Every software-intensive system has a software architecture, whether it is complex or simple. The architecture design is one the earliest and likewise the most relevant artefact that impacts the subsequent artefacts in the systems engineering lifecycle process. Designing the architecture requires a thorough insight in the various quality concerns and their trade-offs [1,9,10]. A systems architecture describes the components of a system, interactions among components, and the interaction of a system as a whole with its environment. A systems architecture is an abstract representation that identifies the higher-level structure of a system and is important for supporting communication among stakeholders, guiding design decisions, supporting the subsequent development process, and analysis of an overall system.

The ISO/IEC 42010 Recommended Practice for Architectural Description of Software-Intensive Systems [6] defines architecture as the fundamental organization of a system embodied in its components, and their relationships to each other and to the environment, and the principles guiding its design and evolution. 

In recent decades, the architecture design discipline has seen the rapid development of applied approaches in architecture modeling, architecture design methods, and software architecture evaluation. With these methods and techniques, the system/software architect makes a wide range of design decisions that lead to the selection of a particular design alternative. In this paper we focus on the design of architectures, and herewith architecture design patterns for digital twins, which we elaborate on in the next sub-section. 

### 2.3. Architecture Design Patterns

Solving a problem usually does not require the development of a new solution, and the provided solution is often not completely distinct from earlier proven solutions. An expert recalls earlier similar problems and subsequently tries to reuse the proven solution. This combination of recurring problems and generic solution templates for a given context is common to many different domains including systems architecture design. Patterns help to build on the collective experience of experts in that domain and capture existing well-proven experience. As such, they help to promote good design practice. Design patterns play an important role in the engineering design process and can be applied at different levels in the lifecycle including architecture design, detailed design, and the code.

The idea of architectural patterns and recording design knowledge in a reusable and canonical form can be traced to Christopher Alexander [11] who proposed the adoption of patterns in the context of the architecture of buildings. In his book *The Timeless Way of Building* he identifies a catalog of patterns that can be applied for designing buildings, neighborhoods, and even whole cities. The underlying paradigm of Alexander is to design so-called living places that do not only fulfil functional requirements but also address important quality factors that imbue buildings with comfort and beauty.

The notion of patterns was subsequently applied to software engineering, as initiated by Gamma et al. with their seminal book *Design Patterns—Elements of Reusable Object-Oriented Software* [12]. The patterns in this book leaned heavily on Alexander’s style to describe patterns but were focused on object-oriented software development. Architectural patterns are similar to software design patterns but have a broader scope. An architectural pattern is a general, reusable solution to a commonly occurring problem in software architecture within a given context [13].

The relationship among the above topics of systems engineering, architectural design, and architectural design patterns is shown in Figure 2. The figure shows that the development of digital twin-based systems is, in essence, a systems engineering activity. Like any system, a digital twin-based system has an architecture that is designed using architectural patterns. In our study we present a catalog of system architecture digital twin patterns. These patterns are used in and related to the lifecycle stages that we presented in Figure 1.

## 3. Research Methodology

We adopted a systematic research methodology to derive the architecture design patterns for digital twins. The key steps are shown in the workflow in Figure 3.

In the first step we provide a thorough analysis of the digital twin literature to derive the key concepts, and the common and variant properties and scenarios. The concept of digital twins is relatively new and as such designing systems based on digital twins is not yet mature. Because digital twins build on the notion of control systems, the domain analysis process also focused on the control systems domain as it is observed, for example, in production control systems. The result of the domain analysis process is a domain model that defines the relationships among the key concepts.

The domain analysis process is followed by the design and analysis of multiple case studies in which digital twins were applied. In this analysis we identify where digital twins have been applied or aimed to be applied. In fact, this step aims to discover patterns rather than inventing these from scratch. The case study was carried out as part of the European IoF2020 project in close interaction with the involved business partners [14]. The project included 19 IoT use cases that were organized in five coherent trials that aim to address the most relevant challenges for the concerned sub-sector [15]. The use case architectures were modelled based on an architecture framework of IoT-based food and farm systems [16]. Most of these case studies address the implementation of digital twins but implicitly and in different ways. We selected three cases for illustrating each of the digital twin patterns. The detailed descriptions are shown in Table 1.

Using the above three case studies, expert knowledge was distilled from the proven solutions which were compared to the domain model and analyzed, and then explicitly documented in a reusable format. An additional input to the pattern identification and documentation process is the literature on existing software and systems design patterns. Some of the identified digital twin patterns build on or resemble existing design patterns. The remainder of the paper introduces the results following the research steps as described above.

## 4. Domain Analysis of Digital Twins

Domain analysis can be defined as the process of identifying, capturing, and organizing domain knowledge about the problem domain with the purpose of making it reusable when creating new systems. The term domain is defined as an area of knowledge or activity characterized by a set of concepts and terminology understood by practitioners in that area. Conventional domain analysis methods consist generally of the activities Domain Scoping and Domain Modeling.

### 4.1. Domain Scoping

Domain Scoping identifies the domains of interest, the stakeholders, and their goals, and defines the scope of the domain. An important activity hereby is the identification of the primary studies from which the concepts are derived. There are now an increasing number of publications on digital twins. We have selected the studies that explicitly discuss the design of digital twins and that were pioneering. The selected list of primary studies that we used to analyze the concept of digital twins and, in particular, the design of digital twins are as follows: [1,14,17,18,19,20,21,22]. 

### 4.2. Key Concepts

In our domain analysis process, we focused on definitions, concepts, and design of digital twins. A digital twin is basically a digital replica of a real-life object to which it is remotely connected [1,23]. The domain scoping is followed by the domain modeling process which is the activity for representing the domain or the domain model. Typically a domain model is formed through a commonality and variability analysis to concepts in the domain.

The concept of digital twins has increasingly become an area of interest to support the development and analysis of smart systems. Various studies in the literature define a digital twin. However, it is also worthwhile to indicate what a digital twin is not, that is, the misconceptions about the term, which have also been discussed in the literature. The various views and misconceptions about the digital twin concept is shown in Figure 4. In common with all of these views is that they distinguish a digital object from a physical object, while indicating there is a relationship between these. There seems to be an agreement on the notion of digital object and physical object, but the semantics for the relationship are different. The relationship between the digital object and physical object may or may not be automatic [1]. In addition, in the list of definitions presented later, only the case in which a digital object is automatically related to the physical object is relevant to the notion of digital twins. Ideas beyond these automated and synchronized states of both objects are considered to be misconceptions in the strict sense. In the first view of Figure 4, digital model, digital object, and physical object are loosely connected and the synchronization or data flow between these occurs through manual intervention. There is no automated translation or interpretation between both objects. In the second view, the digital generator, a digital model is used to automatically generate or enhance a physical object. Thus, generation techniques as defined in the model-driven development could be used. In this alternative, the dataflow from physical object to digital object is missing or is based on manual intervention only. In the case of the digital shadow, mechanisms are provided (e.g., sensors) to provide an automatic data flow to the digital object. This could be needed for analysis or simulation purposes. In the last alternative, digital twin, the digital object and physical object are causally connected and synchronized.

An analysis of the literature leads to useful insights into the notion of digital twins. Jones et al. [22] provided a common list of these concepts based on a systematic literature review. Based on these terms a conceptual model was provided. The key terms that we use in this paper are shown in Table 2. In addition, authors have identified several open research issues related to digital twins. An aspect that is not explicitly considered, however, is the notion of reuse-based design of digital twins.

The application of digital twins is in essence a control system activity to enhance the intelligence of the system and the overall decision-making process herein. As such, the notions of control and smart systems appear to be essential. A model for a control system is shown in Figure 5 (based on [1,24,25]). In the figure, the rectangles represent control entities, while the arrows represent control flows.

The capabilities of smart systems can be grouped into four areas: monitoring, control, optimization, and autonomy [25,26]. Each capability builds on the preceding one: control requires monitoring, optimization requires control and monitoring, and autonomy requires all three. Monitoring implies the observation of a system’s condition, operation, and external environment through sensors and external data sources. Control implies the regulation of systems through remote commands or algorithms that are built into the device or exist in the cloud. The data collected from the monitoring activity together with the control capability allows the optimization of the system. Optimization can be implemented using dedicated algorithms and analytics that can adopt the monitoring data optionally with the historical data. The goal of optimization is typically to improve the quality of the system including its effectiveness and efficiency. The highest level of smart behavior is autonomy, which combines monitoring, control, and optimization to learn about the environment, self-diagnose the own goals and needs, and adapt to the changing preferences. An autonomous system may be interfaced with other systems but controls its own actions.

Based on the higher-level model of the digital twin Figure 4 and the control model of Figure 5, we now provide the digital twin model as shown in Figure 6. In essence, a digital twin is thus considered a control system, but the control is now carried out in the digital object space. The model itself will be used to illustrate the digital twin patterns in the following sections.

## 5. Architecture Design Patterns for Digital Twins

The conceptual model of Figure 6 itself is not sufficient to provide the system architecture of a digital twin-based system. The model by itself is generic and can be used to support the understanding of various systems. However, no detail is provided about the type of digital twin or the required design. Earlier, we indicated that different types of smart systems could be distinguished including monitoring system, control system, and autonomous system. In alignment with and related to this observation we provide different design patterns that are useful for designing a digital twin-based system.

A design pattern is understandable and reusable if it is well-documented. In general, the documentation for a design pattern includes the context in which the pattern is used, the recurring problem description within the context that the pattern seeks to resolve, and the suggested solution template. Unfortunately, there is no single, standard format for documenting design patterns, but rather a variety of different formats have been proposed. Inspired by the documentation templates as used in [10,12], in this paper we adopt the documentation template as shown in Table 3. In alignment with the conventional pattern documentation approaches, the catalog of architecture design patterns does not specify a specific system architecture. Rather design patterns function as templates that need to be instantiated for the particular context of the system. A systems architecture might use one or more of these patterns.

In the following subsections we use this documentation template to describe the identified digital twin patterns, which can be applied in the different stages of the systems engineering lifecycle. Thus, we cover a complete lifecycle, which also means that patterns are addressed that do not include two-way synchronization (Figure 4), i.e., a Digital Model, Digital Generator, Digital Shadow, and Digital Proxy pattern. The “real” digital twin patterns that we identified include a Digital Monitor, Digital Control, and Digital Autonomy. For each of these patterns we also present the possible variations leading to a comprehensive pattern catalog.

### 5.1. Digital Model Pattern

The first pattern we discuss is the Digital Model pattern as shown in Figure 7. This pattern is related to the concept and development stage of the systems engineering lifecycle. In essence it does not yet result in a truly digital twin-based system due to the manual creation and the lack of continuous synchronization. However, it is regarded as the initial step towards a digital twin-based system. Based on a predefined digital model (design), a physical object is developed.

### 5.2. Digital Generator Pattern

The Digital Generator pattern (Figure 8) is similar to the Digital Model pattern in that it defines the production of a physical object based on a digital model blueprint. The difference is that the production is now automated and human intervention is thus not needed. The implementation of this pattern requires therefore also the implementation of the generation process that takes as an input the digital blueprint and provides as an output a physical object or part of the physical object. The automated production of the physical object could be based on techniques as defined in model-driven development.

### 5.3. Digital Shadow Pattern

The Digital Shadow pattern (Figure 9) provides a digital twin based on the physical object. Hence, it assumes already the existence of a physical object. Multiple different digital twins can be generated for one or more physical objects, based on the defined system requirements. The implementation of this pattern requires that the physical object is equipped with the necessary sensors to derive the information that needs to be modeled by the digital twin. In principle, the development of a digital twin based on a physical object could be also a manual process, which would be considered a kind of manual reverse engineering process.

### 5.4. Digital Matching Pattern

The previous three patterns focused on the creation of a physical object based on a digital twin (Digital Model, Digital Generator), or vice versa (Digital Shadow). The creation is either manual or automatic. In the Digital Matching pattern we assume an existing digital model that is used to find physical objects that match the properties of the digital model. The addressed problem is thus not creation, but search (physical object) and match (with digital twin). Typical applications of this pattern are, for example, classification and object recognition applications using machine learning or deep learning approaches. The pattern is shown in Figure 10. This pattern is related to the utilization and support stages of the systems engineering lifecycle. The matching process is assumed to be automatic. Therefore, this pattern requires the implementation of search, classification, and object recognition algorithms.

### 5.5. Digital Proxy Pattern

The Digital Proxy pattern (Figure 11) is also a pattern that is applied in the utilization and support stages. The pattern provides a digital twin that is used as a proxy for the physical object. This means that all the communication directed to the physical object is captured by the digital twin, which acts and responds in the name of the physical object. Hence, the digital twin can provide pre-processing and post-processing of the required services. Multiple different digital twins can be developed based on the different needs for communication, pre-processing and post-processing.

Introducing such a digital twin as a placeholder can serve many purposes, including enhanced efficiency, easier access and protection from unauthorized access. To provide the proper behavior, the digital twin that acts as a proxy can include, that is, reflect the state of, the physical object or be stateless and rely on the retrieval of the state of the actual physical object.

### 5.6. Digital Restoration Pattern

The Digital Restoration pattern (Figure 12) is applied when the physical object needs to restore its earlier state. This could be needed due to failures, loss, corruption, or to align with the state of other physical objects and the environment state. The state of the physical object is reflected on the digital twin at defined times. In the case in which restoration is needed, the process is reversed and the state of the digital twin is used to restore the state of the physical object. The pattern is applied in the utilization and support stages.

### 5.7. Digital Monitor Pattern

The Digital Monitor pattern (Figure 13) is used to provide a digital twin that monitors a physical object. One digital twin can in essence monitor multiple physical objects, and a physical object can also be monitored by multiple twins. Hence, the relationship between digital twin and physical object is many-to-many.

### 5.8. Digital Control Pattern

The Digital Control pattern (Figure 14) builds on the earlier pattern that monitors the physical object. However, it adds an additional level of intelligence and complexity which is that of control. Based on defined control parameters and a comparison mechanism, the proper action is decided which is then performed on the physical object. This pattern thus follows the behavior of a smart feedback control system. Its implementation requires the development of sensors, definition of control parameters, implementation of the comparison algorithms and the decision module, and finally the actuator mechanism. The control parameters are typically externally defined by the client. The decision making for the proper action can be guided by the accompanying data analytics.

### 5.9. Digital Autonomy Pattern

The Autonomous Autonomy pattern (Figure 15) builds even further on the Digital Control pattern and does not require manual, human intervention. The control parameters are now defined by the digital twin itself and based on the changing conditions, if needed, these are adjusted. As a result, the digital twin does not just follow earlier defined control actions but is also able to learn from the dynamically changing situation, and thus adapt its control actions.

### 5.10. Pattern Selection Approach

In the previous section we presented the pattern catalog including nine different design patterns for digital twins. Once these patterns have been explicitly described, each of these can be selected based on a recurring problem, and applied to derive a design based on the pattern. The applicability of these patterns is defined by the systems engineering stage for which it is mentioned, and of course the identified problem that requires a solution. Figure 16 shows the systems engineering life cycle stages and the potential digital twin patterns that can be applied at these stages. Note that the patterns Digital Model, Digital Shadow, and Digital Generator can be applied in the earlier stages of the lifecycle. The remaining patterns are applied in the later stages. The Digital Proxy pattern can also be applied in the retirement stage to behave as a representative of the physical object, even if this has been disposed of.

## 6. Case Study Research

To illustrate the application of the digital twin patterns we adopted a multiple case study approach. As stated before, the case study was carried out as part of the European IoF2020 project in close interaction with the involved business partners (www.iof2020.eu). The primary objective of the case studies is to evaluate the applicability of the identified digital twin patterns. To evaluate the above research questions, the case study research protocol as defined by Runeson and Höst [27] was applied. Based on this, the indicated five steps were followed: (1) case study design; (2) preparation for data collection; (3) execution of data collection on the studied case; (4) analysis of collected data; (5) reporting.

The case study research was applied to three selected case studies. Each of these studies were prospective cases, that is, they included the system that was planned to be developed. The detailed descriptions of these case studies were provided in Table 1.

Table 4 shows the case study design elements. The goal for the case study was to evaluate both the effectiveness and the practicality of the approach. The research questions were defined accordingly as shown in the table. For the adopted background and sources in the case study research, official design documents were used and unstructured interviews were conducted with project managers and system architects. A qualitative data analysis approach was used in which document analysis approaches were used.

Table 5 shows the result of the overall case study research with the identified patterns for each of the three case studies. A number of interesting observations can be derived from the table.

The Digital Model pattern can only be applied for non-living things, in this case the greenhouses. The application of the Digital Generator pattern is different for different case studies. For example, a cow as a physical object is different from a greenhouse. In the case of a greenhouse, the Digital Generator can make sense, whereas for a cow this is not possible. The generation often involves the generation of the computational modules. For a physical generation, this would imply techniques such as 3D printing of physical objects. The Digital Shadow pattern is broadly applicable since it assumes a capturing of the state of a physical object. Hereby, the nature of the physical object is less decisive. However, this will impact the implementation of the pattern because different types of sensors might be needed for different types of physical object (field, cow, greenhouse). The Digital Restoration pattern was planned to be used for the greenhouse case study, to undo/redo facilities to restore/update the state of greenhouses (temperature sensor, light sensor, humidity sensor, and actuators). Again, this pattern can be typically applied for non-living things. The Digital Monitor and Digital Control patterns appeared to be applied in each of the three case studies, although what was monitored and how (sensors) and what was adapted (actuators) was clearly different per case study. None of the case studies used the Digital Autonomy pattern. This is indeed the most difficult pattern, and requires sophisticated knowledge and implementation. We foresee that this pattern can be used in the near future.

## 7. Conclusions

The notion of digital twins can be considered as a new phase in IoT-based systems that will have an increasing and lasting impact on many application domains. It enables remote control of operations based on (near) real-time digital information instead of having to rely on direct observation and manual tasks on-site. Despite its pervasiveness and increased popularity, there appears to be a lack of understanding and consensus on both the basic concepts and, in particular, the design and modeling abstractions. Developing digital twin-based systems requires a systems engineering approach due to its multidisciplinary nature. One of the essential concepts in systems design is the notion of design patterns, which has been largely applied in software engineering and, more recently, also in the broader systems engineering context. Based on our study, this article is the first that proposes a design patterns catalog that can be used to leverage the development of high quality digital twin-based systems. The design patterns catalog is based on a conceptual model of control systems and includes a total of nine different design patterns that address different problems and that can be applied to different systems engineering life cycle stages. The case studies show the application of these patterns in the agriculture and food domain. The usage of digital twin patterns was explicitly identified in cases that were not yet framed as digital twins. The identified patterns focus on the usage stage of the life cycle. The application of patterns in the concept, realization, and retirement stages was still in an early stage of development. We analyzed and checked the digital twin patterns for retrospective case studies that showed the applicability of these patterns. In addition to the food and farm domain, the proposed pattern catalog can be used for various systems engineering applications. In our future work we will focus on the implementation of the presented digital twins, and also consider multiple different case studies from multiple different systems engineering domains. The exploration of other domains will help to discover additional digital twin patterns and thus extend the pattern catalog.

## Figures and Tables

**Figure 1 sensors-20-05103-f001:**
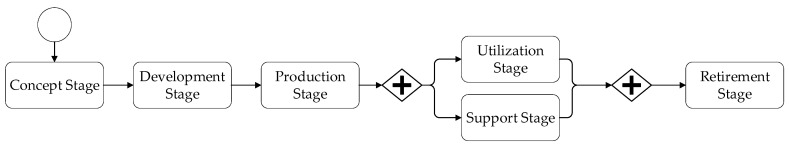
ISO/IEC 15288 System Lifecycle with the key lifecycle stages [9].

**Figure 2 sensors-20-05103-f002:**
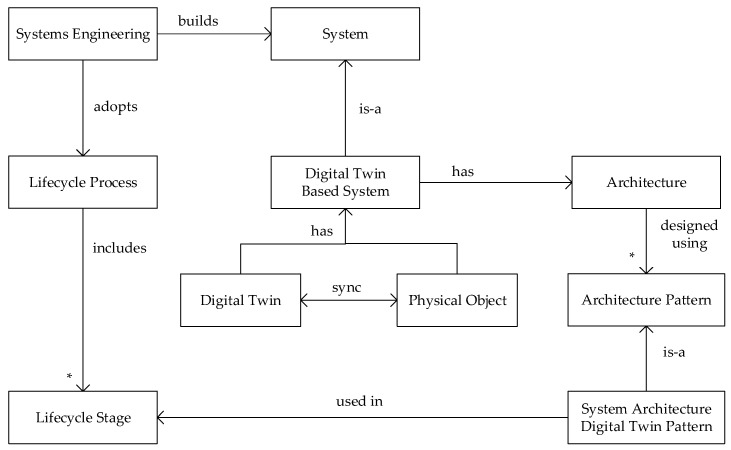
Metamodel for system architecture digital twin patterns.

**Figure 3 sensors-20-05103-f003:**
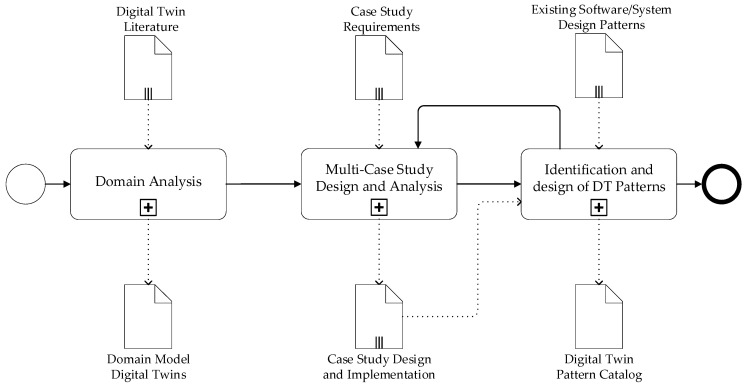
Adopted research methodology.

**Figure 4 sensors-20-05103-f004:**
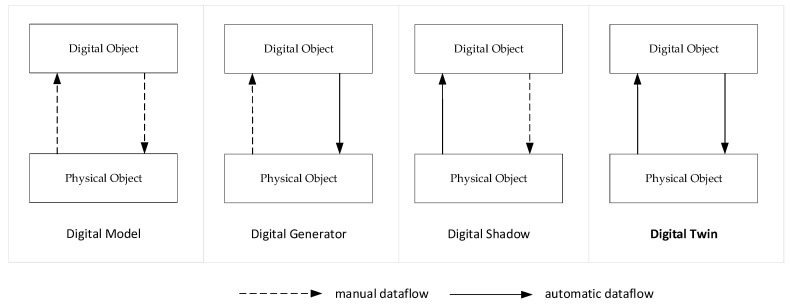
Identified relationships between digital object and physical object.

**Figure 5 sensors-20-05103-f005:**
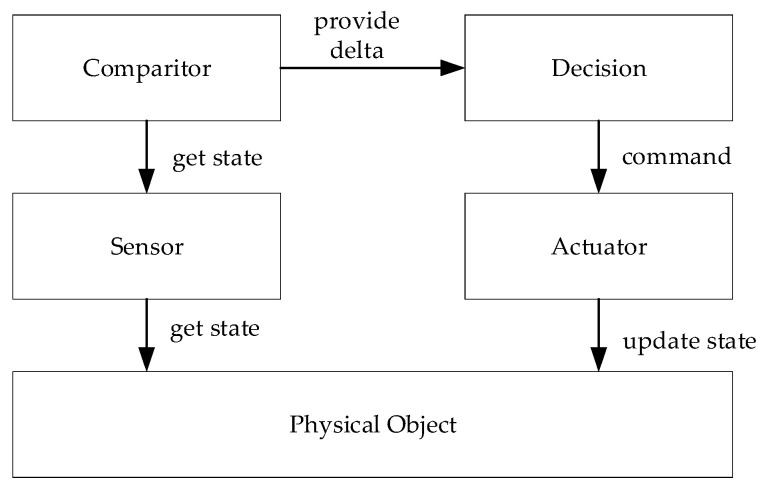
Conceptual model for a control system as derived from the literature.

**Figure 6 sensors-20-05103-f006:**
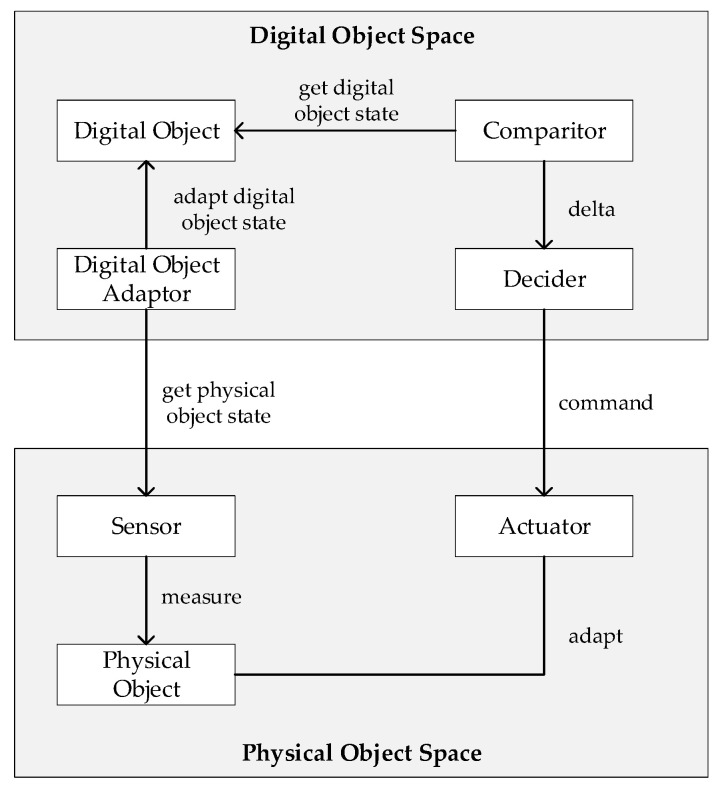
A conceptual model for control-based digital twin as derived from the literature.

**Figure 7 sensors-20-05103-f007:**
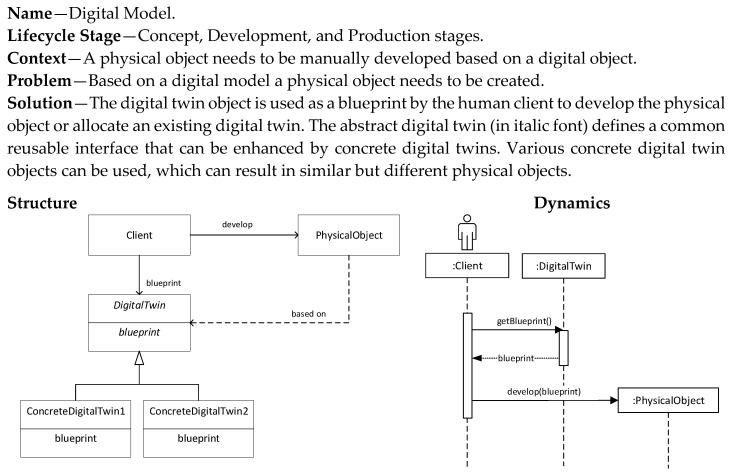
Digital Model pattern.

**Figure 8 sensors-20-05103-f008:**
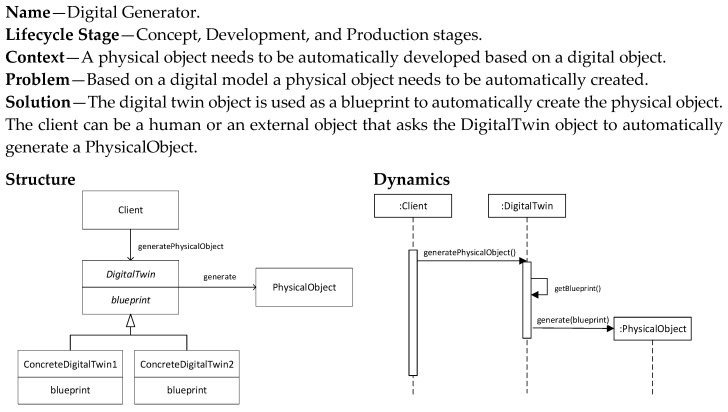
Digital Generator pattern.

**Figure 9 sensors-20-05103-f009:**
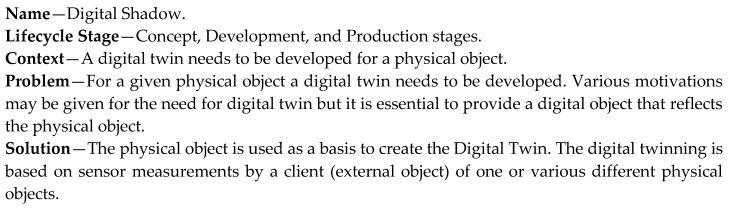
Digital Shadow pattern.

**Figure 10 sensors-20-05103-f010:**
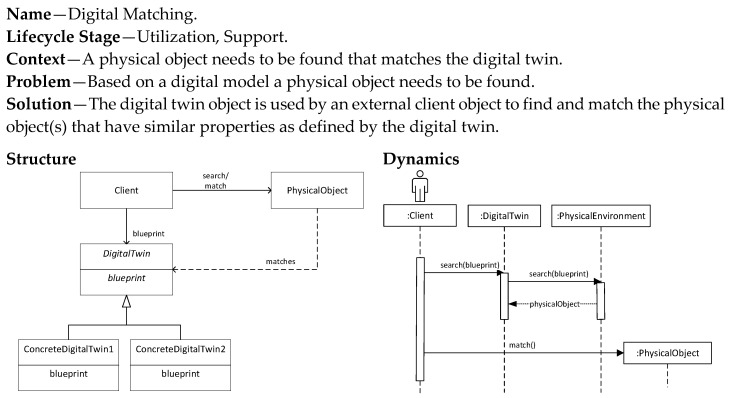
Digital Matching pattern.

**Figure 11 sensors-20-05103-f011:**
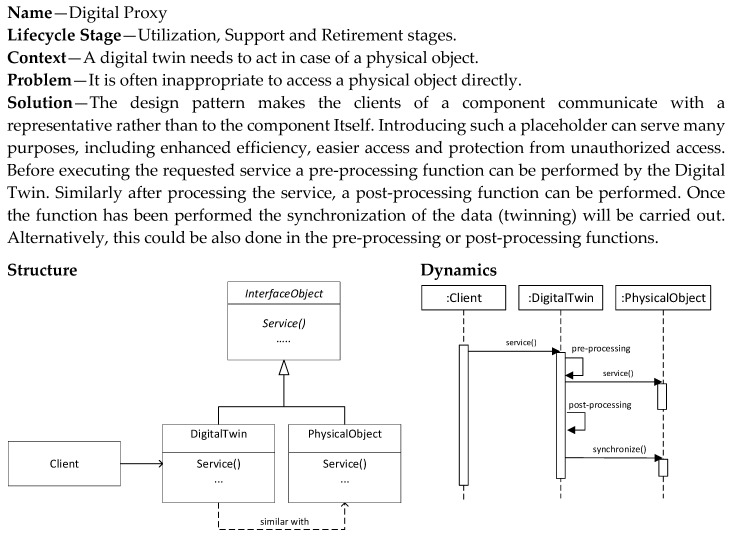
Digital Proxy Pattern.

**Figure 12 sensors-20-05103-f012:**
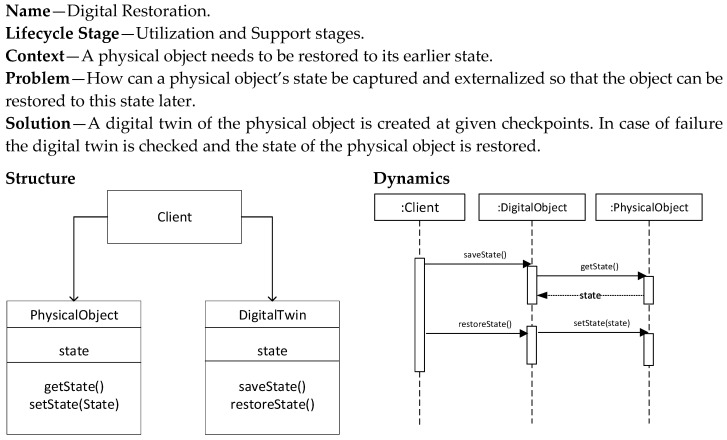
Digital Restoration pattern.

**Figure 13 sensors-20-05103-f013:**
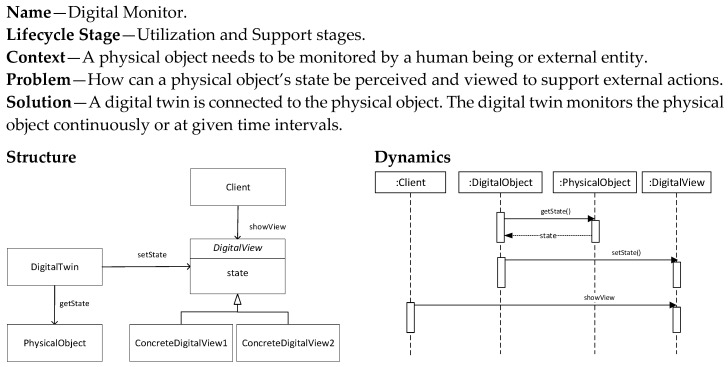
Digital Monitor pattern.

**Figure 14 sensors-20-05103-f014:**
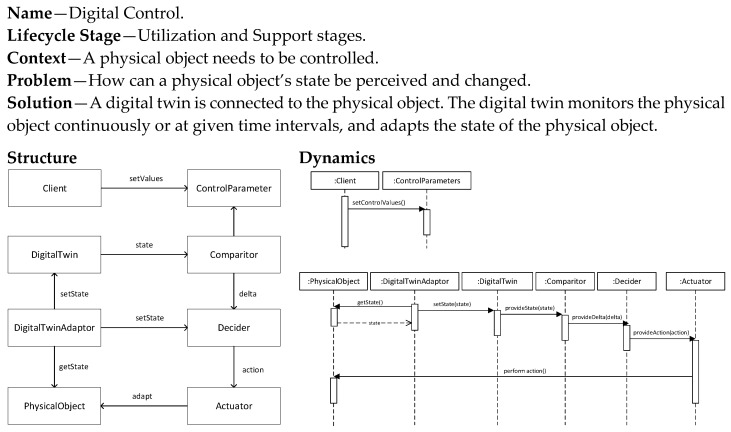
Digital Control Pattern.

**Figure 15 sensors-20-05103-f015:**
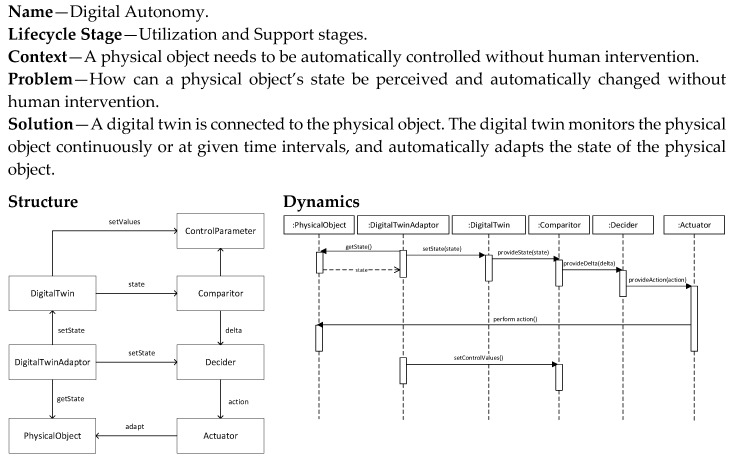
Digital Autonomy pattern.

**Figure 16 sensors-20-05103-f016:**
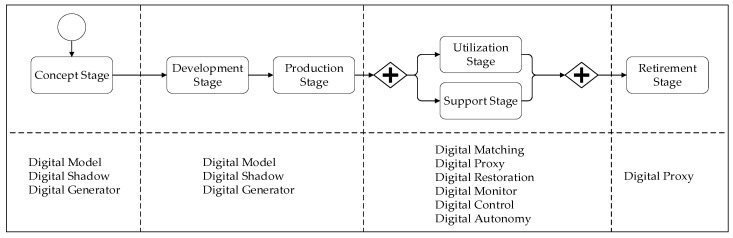
Identified patterns in the systems engineering life cycle stages.

**Table 1 sensors-20-05103-t001:** Overview of the adopted use cases.

Trial/Sector	Use Case	Description
Arable	Within-field management zoning	This case study focuses on within field management zoning and precision farming in arable farming. Hereby with the use of sensors, connectivity, decision support tools and smart control equipment are used to capture and transmit geo-localized real-time information at low cost. The collected data from the sensors will be processed and analyzed to measure and monitor the state of the agro-environment, e.g., soil, crop and climate. Further, the data and the decision models will be combined with agro-climatic and economic models, forecasts and advices for supporting tactical decisions and operational management of technical interventions.
Dairy	Happy Cow	The case study aims to improve dairy farm productivity using 3D cow activity sensing and machine learning techniques. Using advanced sensor technology within farm management it is aimed to monitor the cow behaviors and provide predictive analytics to provide insight on heat detection and health, and thereby support in the decision-making process and recommend feasible solutions to farmers. Data is gathered at both the cow level and herd level, to understand both individual animal and herd characteristics. For different problems, different types of sensors are used which are located, for example, in the neck of the cows (more comfortable position) during daily activity. The collected data during the day is transmitted through a high-efficient, long-range wireless communication network and stored on the cloud for the data analytics and decision-making process.
Vegs	Chain-integrated greenhouse production	The focus of this case study is on developing IoT-based greenhouses involving a large amount of data, physical and virtual sensors, models, and algorithms focusing on important aspects such as water and energy use efficiency, safety, and transparency, for both conventional and organic supply chain traceability systems of tomato. In this context, the chain-integrated greenhouse production use case aims to develop a decision support system (DSS) for the greenhouse tomato supply chain based on IoT technology and the digital twin concept. With an integrated approach based on standardized information, interoperability along the production chain will be increased. This in turn will support quality and safety management, improved products and processes, and a lower environmental impact.

**Table 2 sensors-20-05103-t002:** Selected and used key concepts related to digital twins (adapted from [22]).

Term	Description
Physical Object	A “real-world” artefact, e.g., a vehicle, component, product, system, model.
Virtual Object	A digital representation of the physical object.
Physical Environment	The environment within which the physical object exists.
Virtual Environment	The digital environment in which the digital twins of the physical object exists.
State	The current value of all parameters of either the physical or virtual object/environment.
Realization	The act of changing the state of the physical/virtual object/twin.
Metrology	The act of measuring the state of the physical/virtual object/twin.
Twinning	The act of synchronizing the states of the physical and virtual object/twin.
Twinning Rate	The rate at which twinning occurs.
Physical-to-Virtual Connection/Twinning	The connection from the physical to the virtual environment.
Virtual-to-Physical Connection/Twinning;	The connection from the virtual to the physical environment.

**Table 3 sensors-20-05103-t003:** Documentation template for patterns.

Documentation Item	Description
*Name*	A descriptive and unique name that helps in identifying and referring to the pattern.
*Lifecycle Stage*	The lifecycle stage(s) in which the pattern can be applied.
*Context*	The situations in which the pattern may apply.
*Problem*	The problem the pattern addresses, including a discussion of its associated forces.
*Solution*	The fundamental solution principle underlying the pattern.
*Structure*	A detailed specification of the structural aspects of the pattern.
*Dynamics*	Scenarios describing the run-time behavior of the pattern.

**Table 4 sensors-20-05103-t004:** Case study design.

Case Study Design Activity	Case Study
*Goal*	Assessing the effectiveness of the method Assessing the practicality of the method
*Research Questions*	RQ1. To which extent do the defined digital twin patterns support the system architecture design? RQ2. How practical is the method for applying the digital twin patterns?
*Background and source*	Official requirements documents Project managers and system architects
*Data Collection*	Indirect data collection and direct data collection through document analysis and unstructured interviews
*Data Analysis*	Qualitative data analysis

**Table 5 sensors-20-05103-t005:** Overview of the use cases and the identified digital twin patterns.

Use Case Digital Twin Pattern	Within Field Management Zoning	Happy Cow	Chain-Integrated Greenhouse Production
Digital Model	-	-	Each greenhouse production system can be developed based on a digital model (design)
Digital Generator	-	-	Digital twin could be used to (automatically) generate greenhouse production systems.
Digital Shadow	Initially, a digital model is developed for the fields that are monitored. Later on these digital shadows can become digital twins and the other patterns are applied (e.g., digital monitor, digital control)	Initially, a digital model of a cow is developed that captures the relevant states. Later on these digital shadows can become digital twins and the other patterns are applied (e.g., digital monitor, digital control)	Initially, a digital model of a greenhouse production system is developed that captures the relevant states. Later on these digital shadows can become digital twins and the other patterns are applied (e.g., digital monitor, digital control)
Digital Matching	The pattern can be used to support the analysis and classification of the fields based on defined properties in the digital twin	Properties as defined in the digital twin (e.g., for disease detection) can be used to match with cows.	The pattern can be used to support the analysis and classification of the products in a greenhouse, based on defined properties in the digital twin
Digital Proxy	A digital twin can be used as a proxy to provide information about the fields.	A digital twin can be used as a proxy to provide information about a cow.	A digital twin cane be used as a proxy to provide information about greenhouse production.
Digital Restoration	-	-	Digital model includes undo/redo facilities to restore/update the state of greenhouse production
Digital Monitor	Fields can be monitored by digital twin.	Cows can be monitored by digital twin for various physiological data (temperature, rumen and body activity, pH level).	Greenhouse production systems can be monitored by digital twins.
Digital Control	Based on a sophisticated data analytics decision support, yields are predicted, management zones defined and task maps prepared for farm equipment (e.g., variable application of herbicides, water and fertilizers).	Based on a sophisticated data analytics decision support, monitoring various physiological data (temperature, rumen and body activity, pH level), and a cloud-based server application to provide accurate information for daily operations.	Based on a sophisticated data analytics decision support, yields are predicted and task instructions prepared for greenhouse equipment (e.g., climate, lighting, and irrigation).
Digital Autonomy	-	-	-

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
