# Peer review of "Systems Architecture Design Pattern Catalog for Developing Digital Twins"

_sensors, 2020, doi:10.3390/s20185103_

Round 1

Reviewer 1 Report

Digital twin is a very hot filed in academic and industry. This paper want to propose a pattern for digital twin design. 1. The filed of the method is not reflected in the title. Is it suitable for all industry or just for food and farm systems ? 2. If the method is for food and farm systems, what is the characteristics of these systems? How to evaluate its usability ? 3. There are many literatures for review of DT and many works have the definition of DT. Why the authors only choose the literatures in Table 2? 4. If the Fig4 is misconceptions about DT, what is the correct conceptions? The author also use the concept in Fig4 in the later text. 5. The main idea is about design pattern of DT, i.e how to create DT. Some patterns are not design pattern but using patterns. What is the difference between design and use of DT ? 6. How to automate generate a DT?

Reviewer 2 Report

Overall an interesting and very clearly written paper - essentially an extension of [19,20] in Systems Engineering terminology.  To improve the quality and accessibility for the reader, please address the following points:

  1. The results are only supported by the IoF2020 project. However the case studies are not sufficiently documented to support the findings. Please provide references or more details on the case studies.
  2. The referencing style is inconsistent.  All references should be in the references section at the end of the paper.  Therefore, please include ALL references from Table 2 in the references.  Also from Section 6, the fere4nce to Runeson and Höst (2008) should be in the references.
  3. Figure 15 (Autonomous Digital twin) seems to be identical to Figure 14 (Digital Control).  Please correct, or explain why.
  4. Figure 5 shows only the control flow, not the data flow. This would be better understandable if data flow were included.
  5. Similarly in all patterns, only the control flow is shown. In particular in the message diagrams, the data flow should also be shown.
  6. For better understanding of the patterns, the intention behind the term "Client" should be explained in the text beforehand.
  7. Similarly, the intention of "Concrete Digital Twin" should be explained. (see also next comment)
  8. §5.2 Digital Blueprint: Here the diagram is open to interpretation. What is the scope of the Digital Twin in comparison to the ConcreteDigitalTwins?  e.g. if my Digital Twin is a model of a table: are the dimensions in the Twin or in the Concrete Version? If the l2nd case (as I would expect), then the physical object can only be created by the ConcreteDigital Twin. More explanation needed.
  9. §5.7 Digital Proxy: Where is the syncronisation between the Twin & the object? This is necessary for the effective use of the proxy.
  10. Figure 6: This is a specific control architecture - with a reference (which I assume is in the comparitor). There are other patterns which do not use a reference. & Therefore also other possibiliities for Fig 14. I suggest to say it is "A" control pattern, not "THE" control pattern.
  11. The Digital restoration pattern is only supported by the greenhouse pattern. However without further documentation or explanation, I do not find it convincing.
  12. In the conclusion I would include an indication of potential further work in applying these patterns in other domains, and also study to see if there are other patterns.

Some points where the language / sentence was not understandable:

L185/186: ...expert knowledge is distilled from teh intentify...

L187-188: ... documentation proscess is also the literature...

L283 ...twin can essence monitor...

Reviewer 3 Report

The idea of the manuscript is interesting and useful – the possibility to use already existing templates and experience can reduce time and energy wasted while system’s design. Authors give a good theoretical background, discussing what the Digital Twin concept means and what are the misconceptions about this term. Authors also give a good background of such basic concepts as Systems Engineering and Architecture Design, so that even people far from the topic could understand the idea. However, sometimes maybe some definitions are not needed, because the target audience of the manuscript knows, for example, such basic things as who is the systems engineer or what are the system’s lifecycle stages. The Figure 2 is a very interesting way to conclude the Section 2, it really makes this section more understandable and confirms it's need to be included in the manuscript.

However, authors pretend to create the catalog of architecture design patterns, but the presented diagrams (Figures 7-15) don’t present the system architecture. They are just general graphs depicting automatic control. Architecture should be much more precise, containing components that could be directly implemented to the real-world system.

Case study is not described in details, there are no neither description of the income data, nor any results (according to the goal of the paper, as a result can be taken the designed architecture of the control system based on Digital Twins ). So, the “Case Study Research” Section doesn’t look persuasively.

Moreover, there are some small corrections that should be done:

  • There are some misprints. For example:
    1. In the verse 71, should be “…and finally section 7 concludes the paper”
    2. In the verse 83, there shouldn’t be a dot after the word “disposal”
    3. In the verses 87 and 197, what references should be in the brackets “[ref]”?
    4. In the verse 157, please, remove the article “a” before the phrase “digital twin based systems”, because the word “systems” is plural
  • Table 2 is strangely organized. It is not understandable, why to put this into the table, if this table has only 1 column. The same can be written just in the text. I would suggest either to make this table more informative, or to remove it at all and replace by the text. What is more, all these studies have to be listed in the References section
  • In the Figure 5, “get state” arrows should be opposite. In the Figure 6, “get digital object state”, “get physical object state” and “measure” arrows should be opposite, and “adapt” doesn’t have arrow at all: it should have down arrow. All in all these 2 figures represent control fundamentals, therefore, these figures can’t be presented in the manuscript as conceptual models developed by authors.
  • I suggest to check and correct Figures 7-15. Sequence diagrams should express the sequence of actions in precisely defined time frames. Objects lifecycle timing is the fundamental part of Sequence Diagram and cannot be depicted so freely. Arrow reaching the object's timeline should initiate it. It cannot be initiated before predicting upcoming events.
  • The definition of stakeholder is given in the verse 123, however, this word was used in the manuscript earlier (verses 91, 98, 101, 117)
  • There are 8 self-citations in 23 references, so practically 25% of all references are self-citations. It is too much, I think. References 4 and 5 look as if they were inserted only to be cited, as well as 21 and 23. References 10 and 16 are even not mentioned in the text. I would also suggest to check, how to format references according to the journal’s template, and correct them.

Round 2

Reviewer 1 Report

This paper proposed a category and classification for DT based on system engineering method. The system architecture is interesting. 1. In Fig4, the title maybe not suitable. the fig is about the relationship between digital and physical object. the authors gave them different name. It is not all for "digital twin". If it is correct, maybe it is more reasonable to include the digital matching and all other patterns (line 370-373) in the Fig? 2. Line 444, Figure 7 should be Fig10. 3. Is it design pattern or application pattern ? How to strictly define the design pattern?

Reviewer 3 Report

Authors have tried to answer to all my comments, but still there are some remaining:

  • Authors have transformed Table 2 into the plain text according to the suggestion, but the font size is not the same as in the manuscript. Moreover, the dots should be put after the brackets with references.
  • In the verse 556, the link www.iof2020.eu should be presented either in the footnote or should be added into the reference section
  • Referring to my previous comment about sequence diagrams, I need to stress once again that they should be corrected. Please, don’t mix objects, timelines. Arrows shouldn’t reach objects themselves, just their timelines. Boxes on the timelines symbolize the activation of a given object on the diagram. It cannot be activated before the activation message reaches it either from an external actor or from other objects. Your diagrams are understandable, but are not precise. I suggest to refer to UML 2 standard description, e.g. UML official specification https://www.omg.org/spec/UML/2.5/About-UML/ (chapter 17.2.4).
